# Public Health Implications of Introducing Human Papillomavirus (HPV) Vaccination in Pakistan: A protocol for a mixed-method study to explore community perceptions and health system preparedness

Khola Noreen[1]*, Samina Naeem Khalid[2], Mohsin Javaid[3], Shahzad Ali Khan[4]

**1** Associate Professor, Department of Community Medicine, Rawalpindi Medical University, Rawalpindi, Pakistan, **2** Professor Public Health, Health Services Academy, Islamabad, Pakistan, **3** Demonstrator Health Services Academy, Islamabad, Pakistan, **4** Vice Chancellor Health Services Academy, Islamabad, Pakistan

* khauladr@gmail.com

## Abstract

### Introduction

Cervical cancer is a preventable illness, and early vaccination can serve as a primary prevention strategy. Currently, HPV vaccination has not been introduced at the national level in Pakistan, and the vaccine remains unavailable in most regions. However, efforts are underway to launch the HPV vaccination program soon. For a successful roll-out, it is essential to address circumstantial challenges and mitigate vaccine hesitancy, which stems from a complex interplay of sociocultural and contextual factors. Therefore, this study aims to comprehensively evaluate the multifaceted sociocultural, contextual, and demographic factors influencing the uptake of HPV vaccination at the community level.

### Objectives

1.To assess the current level of knowledge, belief, and factors associated with the acceptability of HPV vaccination among potential vaccine recipients and their parents/caregivers 2. To explore stakeholders' perspectives on the launch of HPV vaccination, considering the dynamics of the local population in the resource-constrained country, Pakistan (OBJ 2). 3.To identify the social and behavioral factors that influence HPV vaccination acceptance and hesitancy within a local community (Punjab) (OBJ 3).

### Method

The ethical approval of the study has been already obtained from the ethical review board of Rawalpindi Medical University (843 IREF/RMU/2024). Data will be collected after obtaining informed written consent from parents and assent from daughters.

**Data availability statement:** No datasets were generated or analyzed during the current study(as it is protocol). All relevant data from this study will be made available upon study completion without any restriction.

**Funding:** The author(s) received no specific funding for this work.

**Competing interests:** The authors have declared that no competing interests exist.

Data collection will start from April 2025 and will be completed in six months. Data compilation and results are expected by December 2025.

A convergent mixed methods design will be used as it will enable the merging of qualitative and quantitative data. Data collection will involve a quantitative phase in which data will be collected from potential vaccine recipients (girls between 9–16 years) and their parents/caregivers to assess the current level of knowledge, belief, and HPV vaccine hesitancy. The qualitative phase aims to explore key stakeholders' perspective on the health system's preparedness and capacity for launch and uptake of HPV vaccination. The quantitative findings will be integrated with the qualitative data via the merging and expanding integration techniques to generate confirmed, expanded, and discordant meta-inferences.

## Discussion

This study will comprehensively identify the multilevel contextual and health system factors that influence HPV vaccine uptake. This study will significantly contribute in field of Public Health by providing a foundational basis of first step of cultural adaptation and validation of BeSD tool specifically for HPV Vaccination.

## Backgrovund and rationale

Cervical cancer has become a serious public health concern due to its high incidence rates and significant impact on women's lives. It is ranked as the fourth most prevalent cause of cancer-related deaths among females. Globally, about 604,000 women received a cervical cancer diagnosis in 2020, and 341,000 of them lost their lives to the illness [1].

Within the WHO Eastern Mediterranean Region (EMRO), cervical cancer is the sixth leading cause of cancer in women. Over 47,500 women in the region lost their lives to cervical cancer in 2020, while an anticipated 89,800 women were diagnosed with the disease [2].

In the WHO South-East Asia Region, cervical cancer poses a pressing public health issue. In 2020, the region reported an estimated 190,874 new cases (32%) and 116,015 deaths (34%), making it the third most commonly diagnosed cancer in the region[3]. A report of a recent systematic review from Pakistan reports an age-specific incidence rate (ASIR) of 7.60 per 100,000 women, with 6166 (95% CI 4833, 8305) new cases of cervical cancer are estimated to occur annually [4].

WHO "Global Strategy for the Elimination of Cervical Cancer" offers a roadmap as "90-70-90" targets for 2030 meaning, "90% of females vaccinated against HPV by the age of 15 years, 70% of women have undergone a high screening test and 90% of women with cervical disease get treated"[5]. A modeling study has shown the impact of the HPV vaccine and cervical cancer screening and its treatment on cervical cancer-related mortality and morbidity. Results showed that 100,000 new cases and 250,000 cervical cancer-related deaths can be prevented [6].

Cervical cancer is a preventable illness, and early vaccination can serve as a primary prevention strategy. Cervical cancer is primarily caused by the human papillomavirus (HPV), a common sexually transmitted infection with a high-risk or oncogenic type. Of the more than 100 HPV varieties, 14 are known to cause cancer [7]. A group of HPV strains causes almost all occurrences of cervical cancer. The most carcinogenic HPV strains, 16 and 18, are reported to be linked to over 71% of cervical cancer cases worldwide [5]. A vaccine is available against these two strains [8].

In Pakistan, two internationally approved HPV vaccines are available: the bivalent Cervarix (marketed by GlaxoSmith-Kline, Pakistan) and the quadrivalent Gardasil (marketed by Merck, Pakistan). Gardasil offers protection against genital warts and cervical cancer by blocking HPV serotypes 6, 11, 16, and 18. Cervarix, on the other hand, works against serotypes 16 and 18. However, these commercially available vaccines are quite expensive. Meanwhile, other nations in South Asia, such as Bhutan, Thailand, Sri Lanka, and the Maldives, have started nationwide HPV immunization programs [9]. At the global level, as of the report of June 2021, 91 countries had launched the HPV Vaccination Program at the national level, and pilot projects are being implemented in an additional 38 countries [10].

Developing nations have significantly reduced the burden of cervical cancer through regular screening with the Pap smears. In Pakistan, unfortunately, uptake of cervical cancer screening is as low as 2% [11]. Moreover, currently the HPV vaccination program has not been launched at the national level, nor is it available in most regions across Pakistan, which puts 68.6 million women over the age of 15 years at risk of HPV infection. According to a prediction model study, approximately 111,000–133,000 cases of cervical cancer might be avoided by 90% annual HPV immunization coverage among girls over 9 years of age [12].

Even though vaccination has been introduced in Pakistan, there is a lack of knowledge and poor uptake among the general population. According to a study conducted on reproductive age women at a Karachi tertiary care facility, as few as 20% of females were aware that HPV vaccines exist, and vaccine uptake is less than 10% [13]. In resource-constrained countries like Pakistan, where screening uptake is low, there is an urgent need to introduce the HPV vaccine as a primary prevention strategy. Ideally, this should be implemented nationwide, utilizing regular, evidence-based strategies and effective immunization methods [9].

Despite the established HPV vaccine efficacy, there are still several barriers, such as high costs, inadequate infrastructure, resource limitations, cultural differences, and a lack of knowledge about the shots. Furthermore, the delivery and administration of vaccines face logistical obstacles in neglected and distant locations with a high incidence of cervical cancer [14]. For the HPV vaccine to be launched successfully in Pakistan, it is crucial to address programmatic issues and vaccine hesitancy due to multifarious sociocultural and contextual factors, especially when it comes to integrating it into the routine immunization program at the national level.

Vaccine hesitancy is a new term and a growing threat to Public Health at the global level [15].WHO's "Strategic Advisory Group on Experts (SAGE) on Immunization" defines vaccine hesitancy as "delay in acceptance or refusal of vaccination despite availability of vaccination services. Vaccine hesitancy is complex and context-specific, varying across time, place, and vaccines. It is influenced by factors such as complacency, convenience, and confidence" [16]. Since Pakistan has faced significant vaccine hesitancy in the past regarding polio [17] and COVID-19 immunizations[18], any negative public perception of HPV vaccinations may lead to catastrophic effects on the rollout of HPV Vaccination at the national level.

Vaccine hesitancy is due to a wide range of complex sociocultural reasons, such as misinformation and myths (such as the idea that vaccines cause infertility), apprehension of side effects, mistrust of the healthcare system, and opinions of powerful leaders [19]. There is an urgent need to address vaccination hesitancy in Pakistan: disinformation, religious misunderstandings, and a lack of confidence in vaccines are the main causes of this rising issue in the context of Pakistan. Because HPV vaccination programs target young girls and teenagers, a group that is heavily impacted by parental decision-making and societal norms, they are more susceptible to these problems. Therefore, it is crucial to assess knowledge, awareness, beliefs, and vaccine hesitancy at the community level. Evidence is required to understand the thinking

behind the decision of parents to approve or reject the HPV vaccine for their daughters and how young girls perceive the launch of the new vaccination. Moreover, as policymakers and national stakeholders are responsible for the launch of vaccination, it is crucial to explore their perceptions and level of preparedness for the HPV vaccine rollout. Similarly, it is imperative to assess the health system's readiness in terms of infrastructure, cold chain management, and supply chain logistics.

There are few studies in Pakistan assessing knowledge [20], attitude [21,22] and preventive practices[23,24] but there is a real dearth of literature regarding the health system perspective on the successful launch of vaccination at the national level. Hence, a health systems approach is imperative to tackle the issue with a holistic approach. To our knowledge, this is the first research to examine HPV vaccination challenges from the perspective of national stakeholders. This mixed-method study will provide a comprehensive and wide-ranging approach to launching a vaccine, encompassing not just the end-user perspective (parents and young girls) but also the lens of health system preparedness.

## Methods

### Conceptual framework

With no one-size-fits-all solution to vaccine hesitancy and acceptance, tailored and context-specific approaches are crucial. The WHO Social Behavior Change Framework (BeSD) [25], displayed in (Fig 1), provides a robust approach for comprehending and altering vaccine-related behaviors. It categorizes the factors influencing vaccine uptake into broadly two main domains, i.e., provider recommendations (motivation) and on-site vaccination (practical issues) related to infrastructure. Motivation is explained by two concepts, namely vaccine demand and hesitancy. Demand is a multifaceted concept that is shaped by human interactions with the infrastructure, health system, and government dynamics, making it most appropriately considered an external variable. Practical issues include external factors that influence vaccine uptake, like access, cost, and quality of health care services. These two domains are further organized into four categories: what people think and feel about vaccines, social processes, motivation to vaccinate, and practical barriers to vaccination.

Uniquely, the BeSD framework is accompanied by both qualitative and quantitative data collection instruments and comprehensive supporting guidance for measuring these drivers. The BeSD framework essentially consists if three of two components: a validated quantitative survey, qualitative interview guides, and corresponding user guidance. The present

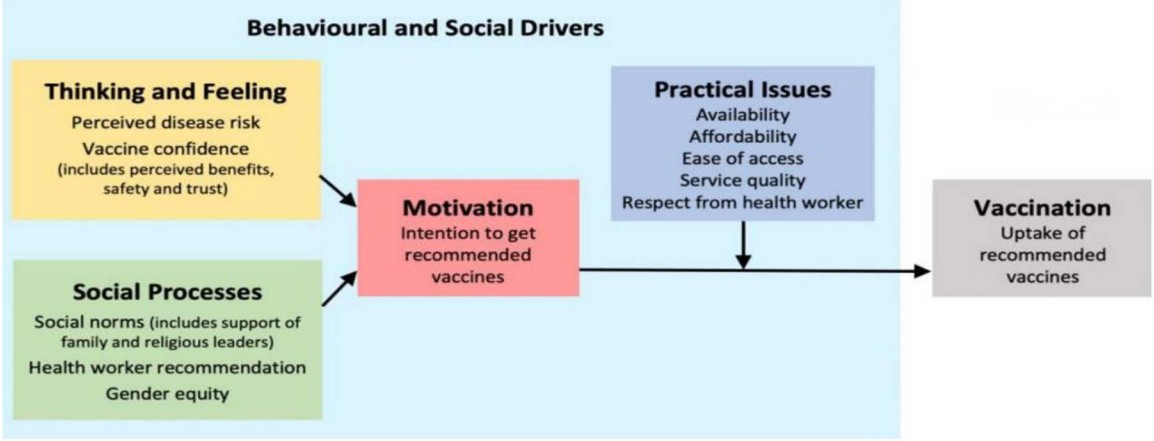

**Fig 1. Behavioural and social drivers of vaccination framework [25].**

study will incorporate elements of the BeSD framework by utilizing its standardized tools, including a validated quantitative survey-based questionnaire and qualitative interview guides for data collection.

### Aim

To understand multifaceted sociocultural and contextual factors influencing the uptake of HPV vaccine to support the launch of HPV Vaccination in Pakistan

### Objectives

1. To assess the current level of knowledge, belief, and factors associated with the acceptability of HPV vaccination among potential vaccine recipients and their parents/caregivers (OBJ 1).

2. To explore stakeholders' perspectives on the launch of HPV vaccination, considering the dynamics of the local population in the resource-constrained country, Pakistan (OBJ 2).

3. To identify the social and behavioral factors that influence HPV vaccination acceptance and hesitancy within a local community (Punjab) (OBJ 3).

## Quantitative phase (OBJ 1 & 3)

### Study design

A cross-sectional survey will be carried out to collect data from potential vaccine recipients including girls between 9–16 years and their parents/caregivers. Data will be collected after six months of approval of the synopsis. The ethical approval of the study has been already obtained from the ethical review board of Rawalpindi Medical University (843 IREF/RMU/2024). Data will be collected after obtaining informed written consent from parents and assent from daughters. Anonymity and confidentiality of study participants will be maintained. Data collection will start from April 2025 and will be completed in six months. Data compilation and results are expected by December 2025. Data will be collected using a self-administered questionnaire.

## Study population and eligibility criteria

### Inclusion criteria

Parents having at least one daughter within the age range of 9–16 years
Girls between the ages of 9–16 years
Only those who give consent after recruitment will form part of the sample

### Exclusion criteria

Parents having daughters already vaccinated against HPV
Having a known contraindication to the HPV vaccine

## Sample size and sampling strategy

### Sample size

The sample size was estimated using the formula ($n = z^2 [p \times q]/ d^2$), where, p is estimated proportion, which in our study was taken approximately as 50%, with margin of error(d) as 5% and formula for q is "$q = 1 − p$ (50%)". The sample size (n) is estimated to be 384. A precise estimated prevalence/proportion ($p$) will not be used because of a lack of studies on the very topic. The sample will be inflated to 500 to enhance generalizability. Data will be collected from 250 parents/caregivers and from 250 potential vaccine recipients (girls between the ages of 9–16 years).

## Sampling strategy

### Data collection from parents and caregivers

The study will be conducted in Punjab Province. This is most populous province of the country having population of approximately 127.69 million people.Cluster random sampling will be employed to ensure maximum representation of diverse population. Cluster sampling will involve dividing population of province into different clusters (districts) then randomly selecting cluster to ensure maximum representation of diverse population across various geographic locations. This approach is chosen to effectively capture a snapshot of current attitudes, beliefs, and behaviors related to HPV vaccination among caregivers in a diverse range of urban, suburban, and rural settings. Participants will be recruited from the general public residing in selected clusters through personal referrals, e-advertisements through social media, approaching the parents of school-going girls, and in the EPI vaccination center of Primary Health Care. The interested participants who are willing to participate and give informed written consent will be screened for eligibility criteria. The parents/caregivers having daughters between 9–16 years will be included as per the Advisory Committee on Immunization Practices (ACIP) recommendation [26].

In order to maximize data collection, data will be collected both physically using hard copies and also using online Google Forms. Considering the sociocultural sensitive nature of the study topic, the option of online data collection will be considered, especially for parents who are reluctant to provide physical data. The questionnaire will be distributed to potential participants using social media platforms like WhatsApp, Facebook, Twitter, and Instagram. Parents will also be approached through personal contacts and by visiting educational institutions and giving questionnaires to schools and university girls to get them filled out by their parents. Additionally, the researchers will reach out to healthcare providers, including nurses, physicians, and EPI staff, to assist in recruiting individuals who may meet the inclusion criteria.

### Data collection from potential vaccine recipients (Girls 9–16 years)

Random sampling will be employed to recruit potential vaccine recipients (girls between the ages of 9–16 years) from educational institutions. The sampling frame will be accessed from the admin office (student section) of the respective institute. Computer-generated random numbers will be used to generate the list of eligible participants. Interested participants will be screened for eligibility criteria based on the inclusion criteria. Justification for the selection of this age group is that the Advisory Committee on Immunization Practices (ACIP) advised adolescent girls to have a routine HPV vaccination at age 11 or 12 years first, then a catch-up immunization at age 13–17 years [26].

## Data collection tools

### HPV vaccination knowledge (OBJ 1)

The "eight-item HPV Knowledge Scale" developed by Karki and colleagues [27] will be used to measure knowledge about HPV vaccination. The "HPV Knowledge Scale" developed by Karki measures information regarding the dose, efficacy, and recommended age and gender for HPV vaccination.

Responses will be recorded on a response scale with options of "True" (score 1) "False" (score 0), and "don't know" (score 0). Since it's an 8-item tool, "total score ranges from 0 to 8", with higher scores indicating greater HPV vaccine knowledge (Attached as supporting document).

### HPV vaccination belief (OBJ 1)

"HPV Vaccination beliefs" will be assessed using the "12-item HPV Belief Scale" by Karki and his colleagues. This scale was modified from an initial version that was developed and validated using a sample of Thai women [28].

The "HPV Belief Scale" constitutes 4 Health Belief Model (HBM) based constructs, namely "perceived susceptibility, perceived severity, perceived benefits, and perceived barriers". There are "two items of perceived susceptibility, three

items of perceived severity, three items of perceived benefits, and four items of perceived barriers in HPV Belief Scale". Responses will be assessed on a 5-point Likert scale ranging from strongly disagree, lowest score (score 1) to strongly agree, highest score (score 5). Items will be summed up, and scores will be calculated by adding responses to each construct. Therefore, perceived benefits scores will range from 3–15, perceived barriers scores will range from 4–20, perceived benefits scores will range from 3–15, and perceived susceptibility range from 2–10. The higher the score, the stronger the belief and vice versa (Supporting document S1 File).

### Behavior and social determinants of HPV vaccination (OBJ 3)

Quantitative data on social and behavioral determinants of vaccine uptake will be collected using a validated survey tool based on the BeSD framework (Attached as supporting document). This framework provides a structured approach to collect data based on social and behavioral drivers of vaccination across 4 domains, namely motivation, thinking and feeling, social processes, and practical issues. The survey questionnaire is divided into 2 sections. The first section includes a sociodemographic profile, including participants' age, marital status, gender, occupation, education level, and vaccination status. The second portion consists of questions related to perceived risk, motivation, vaccination history, social processes, and practical issues influencing vaccine uptake [25].

### Sociodemographic information

Sociodemographic information will include age (respondent age& age of eligible daughter), marital status, gender, occupation, education level, household living arrangement, previous vaccination history, socioeconomic class. Socioeconomic class is categorized as "upper, upper middle, lower middle, upper lower, and lower," modified from Kuppuswamy and Udai Pareekh's scale [29].

## Data collection procedure

### Quantitative data

A web-based application, REDCap, will be used to collect online quantitative data from parents/caregivers and young girls aged 9–16 years. The REDCap is a secure web application for building and managing online surveys and databases. Structured and validated questionnaires embedded in web web-based application will be used to collect data. Internal checks will be done to ensure quality, accuracy, and integrity. Only participants who voluntarily agree to take part in the study will be recruited, with confirmation of their participation obtained through an embedded consent form. Assistance from an IT expert will be sought for the operation of the RedCap application. Additionally, the IT expert will be responsible for conducting training sessions for data collectors, equipping them with the necessary skills for data collection, compilation, and management. This structured approach will help in getting deep insights into vaccine-related knowledge, attitudes, and behaviors, and ensure consistent data collection with minimum bias.

## Data management and analysis

### Socio-demographic variables

For sociodemographic variables, for quantitative variables mean and standard deviations will be calculated. For qualitative variables, frequencies and percentages will be calculated. The "IBM Statistical Package for the Social Sciences (SPSS) Statistics" version 28 will be used for all analyses.

### For HPV knowledge

For HPV knowledge, frequencies will be calculated for correct responses "score of "1" will be assigned to the correct answer, and "0" for "No and Don't know". An individual score of 1–4 will be taken as inadequate knowledge score, while

the score in the range of 11–14 was counted adequate knowledge score. For HPV belief, the responses will be calculated on a 5-point "Likert scale" ranging from "strongly disagree (score = 1) to strongly agree (score = 5). An individual score of < 2.5 is categorized as a low level of belief, and a score of > 2.5 is taken as adequate belief. A chi-square test will be used to calculate the association of the level of knowledge and belief with sociodemographic variables. The p-value of 0.05 will be taken as statistically significant. Logistic regression analysis will be computed with the dependent variable (HPV knowledge) and the set of predictors (sociodemographic variables).

### Behavior and social determinants of HPV vaccination

Quantitative data on social and behavioral determinants of vaccine uptake will be calculated as mean and standard deviation. For HPV vaccine hesitancy, frequencies will be calculated for the correct response. A score of "1" will be assigned to the correct answer, and "0" for "No and Don't Know". A chi-square test will be used to calculate the association of the level of knowledge and sociodemographic variables. The p-value of 0.05 will be taken as statistically significant. Logistic regression analysis will be computed with the dependent variable (HPV) and the set of predictors (sociodemographic variables). For the regression analyses, 5-point Likert scale Scores will be combined into two categories, ranging from "Not at all important" (score = 1) to "Very important" (score = 5). Individual scores of < 3 will be categorized as a low level of hesitancy, and scores of > 3 or more will be categorized as a high level of vaccine hesitancy.

## Qualitative phase (OBJ 2)

### Study design

A qualitative descriptive approach will be used because of its value in obtaining a direct description of phenomena of interest as well as a direct experience from the participants[30].

The strong justification for using a descriptive technique is to give clear accounts of perceptions and feelings, especially in fields where little is known about the topic under investigation and the subject of the inquiry is not well understood [31].

### Study population and eligibility criteria

A sample of national-level key stakeholders and decision-makers will be selected by purposive sampling. The study participants will include national-level key stakeholders and decision-makers, and a few members of the general public.

### Inclusion criteria

Representatives from individual, interpersonal, institutional, and policy levels, including the following:

◦ Policy makers, donors, public health officials, and vaccination committee leads, representatives of country offices of UN agencies (WHO, UNICEF, UNFPA), and international donors and partners (GAVI, Jhepigo); Policy level

◦ Parents of girls 9 to 16 years of age; Interpersonal level

◦ Girls 9 to 16 years of age; Individual level

◦ Healthcare providers; Institutional level

◦ Immunization program leads; Policy level

### Exclusion criteria:

◦ Not willing to participate

◦ Those who don't give informed written consent

## Sample size and sampling strategy

Purposive sampling will be employed as this technique helps to select different stakeholders with diversity among the stakeholder group. Sample size for qualitative data is determined by data saturation, a point where new information no longer contributes to understanding of phenomena of interest, our initial sample size is based on normal saturation points in comparable research [31]. Since point of saturation cannot be predicted in advance so we planned to recruit minimum of 25–30 caregivers and potential recipients, 8–10 policy makers and 25–30 healthcare providers which is in accordance with the typical points of data saturation reported in qualitative studies [30,31]. However we are willing to modify our sample size if needed, and we will continue to recruit until saturation is reached to ensure an indepth knowledge of the factors influencing HPV vaccine uptake.

Those who agree to participate in the interviews will be requested to choose the date and time of the interview as per their feasibility. The interviews will be conducted either in English or Urdu, as per the participants' feasibility. Face-to-face, virtual, in-depth, and key informant interviews will be conducted. Face-to-face interviews will take place at a location of the participant's choice place and virtual interviews will be conducted via Zoom or Microsoft Teams. Interview length will range from 30–60 minutes. Focus group discussion will be conducted to collect data from parents of girls of 9–16 years and girls within the same age bracket. The focus group discussion and interviews will be conducted by two trained research assistants. These assistants will actively interact with the communities by delivering informational leaflets at nearby clinics, hospitals, and schools. Appointments will be made in accordance with the invitations sent to those who show interest in participating.

## Data collection tools

WHO BeSD Qualitative interview guides will be used for data collection (attached as a supporting document S1). BeSD provides a set of four different qualitative interview guides for four different stakeholders, namely potential recipients, healthcare providers, community influencers, and program managers. We will use an interview guide for potential recipients, healthcare providers, and managers [32,33]. Questions can be adapted according to sociocultural context and the research question under investigation.

For qualitative data, Focus Group Discussions (FDGs), indepth interview and key informant interviews (KIIs) will be conducted to collect qualitative data. Since mixed method approach is employed, combination of interviews and focus group discussion will allow thorough comparison of viewpoints, improves the completeness of data, and increases overall trustworthiness by guaranteeing confirmability, dependability, credibility, and transferability of findings. The FGD guide consists of 11 questions focusing on vaccine confidence, perceived risk, social norms, and practical issues related to vaccine access. Probes will be used to supplement each major question in order to provide a more thorough exploration of participants' viewpoints and experiences. Key informant interview (KII) guide tailored to potential stakeholders, containing 11 main questions covering stakeholders' perspectives, recommendations, and potential strategies to promote access to the vaccine. The semi-structured format of these guides allows participants to freely offer in-depth ideas while maintaining consistency throughout interviews, which enhances the data gathered for thematic analysis.Key informant interviews will be conducted with potential stakeholders of EPI, policy makers, donors, public health professionals, and vaccination committee leads, representatives of country offices of UN agencies (WHO, UNICEF, UNFPA), and international donors and partners (GAVI, Jhpiego).Since convergent mixed methods designs involve parallel data collection, both surveys and interviews will be conducted until the required sample is achieved.

## Data analysis plan

Interviews will be transcribed verbatim and entered into NVivo (a software for analysis of qualitative data) for analysis. Data analysis will be conducted using reflective thematic analysis, one of the three approaches to thematic analysis [34]. These three methods differ in "enactment of coding and theme development, underlying research values, and the conceptualization of key concepts" [35,36].

The six-phase process described by Braun and Clarke [37] will be used. Transcripts will be read several times in order to fully comprehend the perspectives of the participants. The trustworthiness criteria will be applied to the qualitative phase to enhance quality. Prolong engagement with the participants, peer debriefing with the team members, and referential adequacy will all help to establish credibility. Moreover, credibility will be achieved through peer debriefing with the research team and referential adequacy by examining the consistency of themes with the raw data. Dependability will be established by keeping an audit trail during data collection and analysis as data analysis. Confirmability will be established by verification of data analysis processes and debriefing with members about the decisions made during data analysis. Transferability will be ensured by a thick description of the sample and context.

## Integration of quantitative and qualitative data

A convergent mixed methods design will be used, involving parallel quantitative and qualitative data collection [38]. Equal weightage will be given to both the qualitative and quantitative phases, as the intention is to merge findings to develop an in-depth understanding. A convergent mixed methods design is pertinent because it will enable the merging of qualitative and quantitative data to comprehensively identify the multilevel contextual and health system factors that influence HPV vaccine uptake (Fig 2).

## Research outcomes and measures

The quantitative data will be integrated with the qualitative data via the merging and expanding integration techniques to generate confirmed, expanded, and discordant metainferences [38]. The quantitative data will be collected using a validated tool, qualitative data collected through study focus group discussion and key informant interviews, and a desk review done by policy review of existing documents will be merged.Thematic analysis will be done to analyze interviews, focus group data, and findings from document review to identify key themes related to policies, practices, and public perceptions. The themes and variable-level data will be presented as integrated results using a joint display (a figure or visual to illustrate how qualitative and quantitative data are integrated). A sample joint display is illustrated as a procedural diagram (Fig 3). This study will significantly contribute in field of Public Health by providing a foundational basis of first step of cultural adaptation and BeSD tool validation specifically for HPV Vaccination.

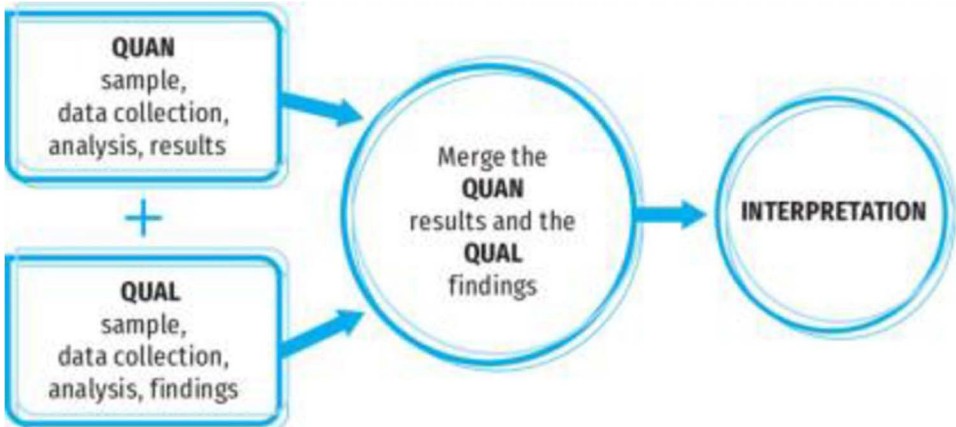

**Fig 2. Convergent Parallel Mixed Methods Design (Plano and Creswell) [36].**

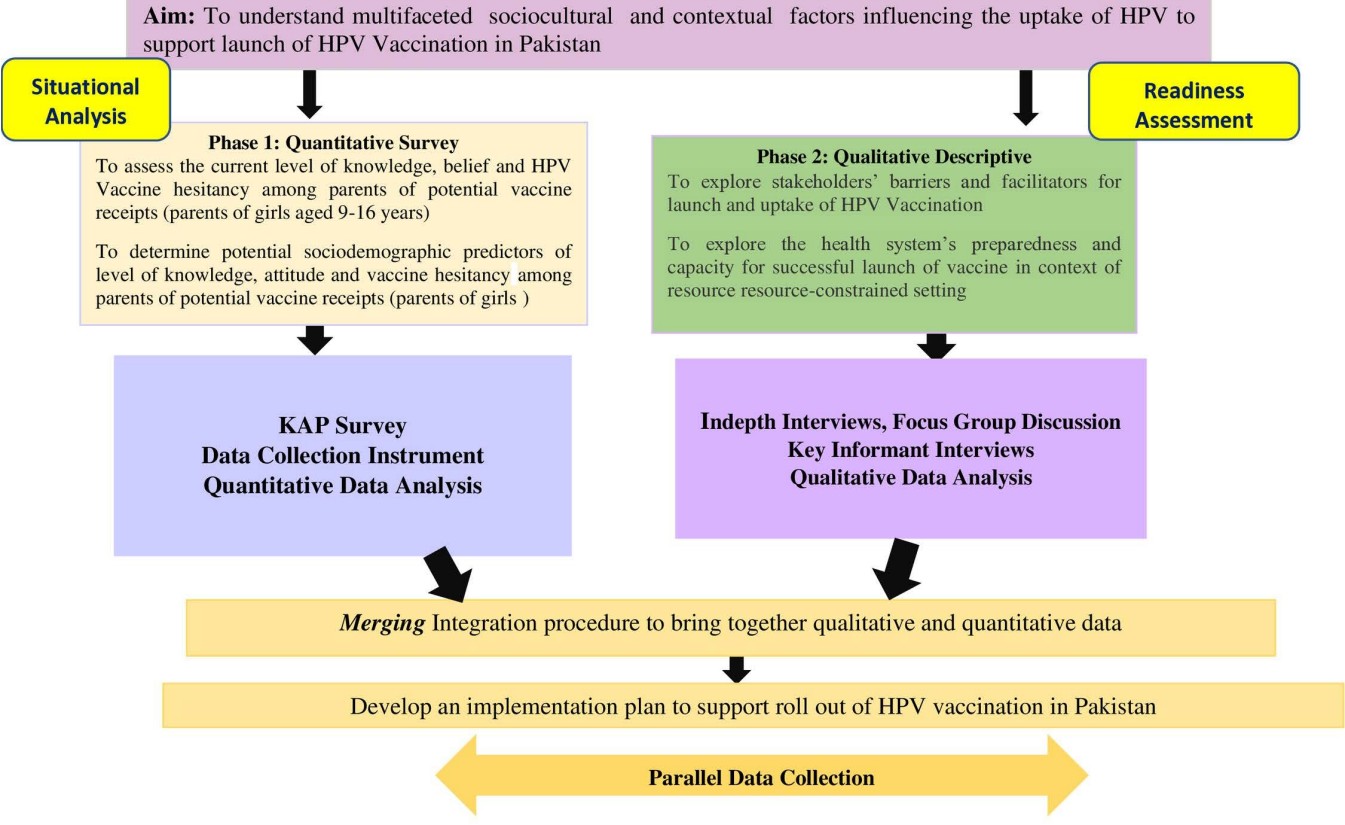

**Fig 3. Procedural Diagram, visual display to show merging of qualitative and quantitative data.**

## Project Management/Governance

### Risk assessment and mitigation

Since it is an observational study, there is minimal risk involved. However, measures will be taken to mitigate any anticipated psychological harm and stigma. We anticipate challenges in recruitment because it is a sociocultural sensitive issue, and many parents will be reluctant to share information. Additionally, parents may not be willing to share their perspective about their daughters. To cater for this issue and considering the privacy, the participants will be given the option of an online platform to provide data with confidentiality and anonymity. This will also address individuals who may have challenges with low socioeconomic status and travel issues due to affordability. Healthcare practitioners, general physicians, EPI Staff, and educational institution staff and faculty will be requested to help with the recruitment.

### Dissemination plan and outputs

**Policy brief**: The findings could inform health policies by highlighting the barriers and facilitators to HPV vaccine uptake in Pakistan. This can lead to more targeted and effective vaccination campaigns.
**Stakeholders' workshops**: To promote opportunities for partnerships and collaboration with stakeholders to support vaccine implementation.
**Community Engagement**: based on community perceptions, beliefs, and attitudes towards vaccination, and develop strategies for effective communication, education, and addressing vaccine hesitancy.

**Journal and Conference Presentation for Academic Audience:** Various passive and active dissemination knowledge translational strategies will be used. The results will be published in international open-access peer-reviewed journals with a wide readership. The findings will also be presented at national, international, and local research symposiums.

**Social Media Campaigns & Press Release**: Social media campaigns on Twitter and Facebook from the webpages of the Public Health at Health Services Academy and Rawalpindi Medical University. The findings will be shared in local newspapers, TV channels, and radio to reach potential knowledge users who rely on these sources for news and those who may not have access to social media platforms and websites.

**Patient Education Session:** One or more educational sessions will be conducted for parents to address their apprehension and demystify the myths and taboos related to vaccines.

## Author contributions

**Conceptualization:** Khola Noreen, Shahzad Ali Khan.

**Methodology:** Khola Noreen.

**Project administration:** Samina Naeem Khalid.

**Software:** Mohsin Javaid.

**Supervision:** Samina Naeem Khalid, Shahzad Ali Khan.

**Writing – original draft:** Khola Noreen.

**Writing – review & editing:** Mohsin Javaid, Shahzad Ali Khan.

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
