## [Decision Letter · Decision Letter 0]

21 Feb 2025

PONE-D-24-59459Public Health Implications of Introducing Human Papillomavirus (HPV) Vaccination in Pakistan: A protocol for mixed method study to explore community perceptions and health system preparednessPLOS ONE

Dear Dr. Noreen,

Thank you for submitting your manuscript to PLOS ONE. After careful consideration, we feel that it has merit but does not fully meet PLOS ONE’s publication criteria as it currently stands. Therefore, we invite you to submit a revised version of the manuscript that addresses the points raised during the review process.

**ACADEMIC EDITOR: **

Edit the content of the manuscript to correct the numerous grammatical errors.

The proposed sampling is not clear.

Specify the age of adolescents of interest, if it is 9 to 16 years or 9 to 17 years.

Merge the analysis for belief, hesitancy and knowledge to avoid the current repetitions in the manuscript.

We look forward to receiving your revised manuscript.

Kind regards,

Folusho Mubowale Balogun

Academic Editor

PLOS ONE

2. Please ensure that you include a title page within your main document. You should list all authors and all affiliations as per our author instructions and clearly indicate the corresponding author.

4. Please remove your figures from within your manuscript file, leaving only the individual TIFF/EPS image files, uploaded separately. These will be automatically included in the reviewers’ PDF.

Additional Editor Comments:

The manuscript needs to be edited as it presently has many grammatical errors.

The sampling is not clear.

Specify if the age of interest for the adolescent s is 9 to 16 years or 9 to 17 years.

Merge the data analysis for belief, hesitancy and knowledge instead of the current repetitions.

Reviewers' comments:

Reviewer's Responses to Questions

**Comments to the Author**

1. Does the manuscript provide a valid rationale for the proposed study, with clearly identified and justified research questions?

Reviewer #1: Partly

Reviewer #2: Yes

2. Is the protocol technically sound and planned in a manner that will lead to a meaningful outcome and allow testing the stated hypotheses?

Reviewer #1: Partly

Reviewer #2: Partly

3. Is the methodology feasible and described in sufficient detail to allow the work to be replicable?

Reviewer #1: No

Reviewer #2: No

4. Have the authors described where all data underlying the findings will be made available when the study is complete?

Reviewer #1: No

Reviewer #2: No

5. Is the manuscript presented in an intelligible fashion and written in standard English?

Reviewer #1: Yes

Reviewer #2: Yes

6. Review Comments to the Author

You may also provide optional suggestions and comments to authors that they might find helpful in planning their study.

Reviewer #1: * The Aim of the paper is not aligned to the objectives.

* Obj1 does not refer to girls yet in the body 2.3 the girls are in the inclusion criteria. Correct and be consistent.

*Rephrase 2.3 to Only those who give consent after recruitment will form part of the sample.

* 2.4 The sampling strategy is not clear how will random sampling occur on social media platforms, or in schools.

* It confusing how will sample frame be from a university? are there girls aged 6-16 years. Please be consistent with the age group ; elsewhere it is 9-16 and some places 9-17 years.

* A reference is mentioned in the sample size calculation without mention of this reference.

* What do you mean by 'Cast" in 2.5 paragraph, elaborate.

* How will data collection on social media platforms be controlled? there are concerns of duplication or same people answering more than once.

QUALITATIVE PHASE

* Will data collection use IDIs or FGDs or both? it is not clear which one, please delineate and explain where each will be used.

Please correct the methodological errors to make the method replicable for use in other settings.

The protocol and the study will provide valuable contribution to the scientific community.

Reviewer #2: State Data Sharing plan

Revise Methodology to elucidate data gathering process for each stakeholder group (i.e one for parents, one for potential vaccinees, and so on)

7. PLOS authors have the option to publish the peer review history of their article (what does this mean? ). If published, this will include your full peer review and any attached files.

**Do you want your identity to be public for this peer review?** For information about this choice, including consent withdrawal, please see our Privacy Policy .

Reviewer #1: **Yes: ** Prof Yolanda Malele-Kolisa

Reviewer #2: No

---

## [Author Response · Author response to Decision Letter 0]

18 Mar 2025

( PONE-D-24-59459)

Public Health Implications of Introducing Human Papillomavirus (HPV) Vaccination in Pakistan: A protocol for mixed method study to explore community perceptions and health system preparedness

PLOS ONE

Dear editor,

Pls find the modified manuscript. We have incorporated all suggestions recommended by the reviewers’. Following is the point-wise reply to the reviewers' comments.

With best regards,

Dr Noreen

ACADEMIC EDITOR:

Edit the content of the manuscript to correct the numerous grammatical errors.

The proposed sampling is not clear.

Specify the age of adolescents of interest, if it is 9 to 16 years or 9 to 17 years.

Merge the analysis for belief, hesitancy and knowledge to avoid the current repetitions in the manuscript.

Response to Comments :

A rebuttal letter that responds to each point raised by the academic editor and reviewer(s). You should upload this letter as a separate file labeled 'Response to Reviewers'.

Reply : rebuttal letter that responds to each point raised by the academic editor and reviewer is uploaded as separate file

A marked-up copy of your manuscript that highlights changes made to the original version. You should upload this as a separate file labeled 'Revised Manuscript with Track Changes'.

Reply : A marked-up copy of manuscript that highlights changes made to the original version is uploaded as 'Revised Manuscript with Track Changes'.

An unmarked version of your revised paper without tracked changes. You should upload this as a separate file labeled 'Manuscript'.

Reply: unmarked version of your revised paper without tracked changes is uploaded as Manuscript

Reply: Manuscript is prepared according to journal requirement

2. Please ensure that you include a title page within your main document. You should list all authors and all affiliations as per our author instructions and clearly indicate the corresponding author.

Reply : Title page added in main document

Reply : Ethics statement added in method section of manuscript

4.Please remove your figures from within your manuscript file, leaving only the individual TIFF/EPS image files, uploaded separately. These will be automatically included in the reviewers’ PDF.

Reply : Figures removed from main manuscript

Reply :Added accordingly

Additional Editor Comments:

The manuscript needs to be edited as it presently has many grammatical errors.

Reply : Manuscript edited grammatical errors corrected

The sampling is not clear.

Reply: Sampling is revised

Specify if the age of interest for the adolescent s is 9 to 16 years or 9 to 17 years.

Reply :Age is specified 9-16 years

Merge the data analysis for belief, hesitancy and knowledge instead of the current repetitions.

Reply :Data analysis for belief and knowledge is merged

Reviewers' comments:

Reviewer's Responses to Questions

Comments to the Author

1. Does the manuscript provide a valid rationale for the proposed study, with clearly identified and justified research questions?

Reviewer #1: Partly

Reviewer #2: Yes

Reply : Thank you for comment

2. Is the protocol technically sound and planned in a manner that will lead to a meaningful outcome and allow testing the stated hypotheses?

Reviewer #1: Partly

Reviewer #2: Partly

Reply : Manuscript is thoroughly revised to further improve the quality

3. Is the methodology feasible and described in sufficient detail to allow the work to be replicable?

Reviewer #1: No

Reviewer #2: No

Reply : Methodology is revised in detail as per reviewers comments

4. Have the authors described where all data underlying the findings will be made available when the study is complete?

Reviewer #1: No

Reviewer #2: No

Reply : Statement is added that all data will be freely available without restriction

5. Is the manuscript presented in an intelligible fashion and written in standard English?

Reviewer #1: Yes

Reviewer #2: Yes

Reply : Thank you for comment

6. Review Comments to the Author

You may also provide optional suggestions and comments to authors that they might find helpful in planning their study.

Reviewer #1: * The Aim of the paper is not aligned to the objectives.

Reply : Aim is revised

Reply: Aim is revised “To understand multifaceted sociocultural and contextual factors influencing the uptake of HPV to support launch of HPV Vaccination in Pakistan “

* Obj1 does not refer to girls yet in the body 2.3 the girls are in the inclusion criteria. Correct and be consistent.

Reply :Data will be collected for Objective 1 &3 , girls are part of objective 3

*Rephrase 2.3 to Only those who give consent after recruitment will form part of the sample.

Reply :Rephrased accordingly

* 2.4 The sampling strategy is not clear how will random sampling occur on social media platforms, or in schools.

Reply : Sampling strategy is revised and convenience and random sampling mentioned separately

* It confusing how will sample frame be from a university? are there girls aged 6-16 years. Please be consistent with the age group ; elsewhere it is 9-16 and some places 9-17 years.

Reply : Its replaced with educational institution

* A reference is mentioned in the sample size calculation without mention of this reference.

Reply :Since no previous literature avaialable so its taken as 50% . Statement is rephrased to make it more clear and explicit.

* What do you mean by 'Cast" in 2.5 paragraph, elaborate.

Reply :removed

* How will data collection on social media platforms be controlled? there are concerns of duplication or same people answering more than once.

Reply :Data will be collected through REDCap to avoid duplication Details of data collection added

QUALITATIVE PHASE

* Will data collection use IDIs or FGDs or both? it is not clear which one, please delineate and explain where each will be used.

Reply :

Each one explained separately

Please correct the methodological errors to make the method replicable for use in other settings.

The protocol and the study will provide valuable contribution to the scientific community.

Reply : Thank you for your comment , methodological error corrected as per guidance

Reviewer #2: State Data Sharing plan

Reply :Data Sharing plan added

Revise Methodology to elucidate data gathering process for each stakeholder group (i.e one for parents, one for potential vaccinees, and so on)

Reply : data gathering process for each stakeholder group added seperately

---

## [Editor Report · Decision Letter 1]

6 Apr 2025

PONE-D-24-59459R1Public Health Implications of Introducing Human Papillomavirus (HPV) Vaccination in Pakistan: A protocol for mixed method study to explore community perceptions and health system preparedness

PLOS ONE

Dear Dr.Khola Noreen,

Thank you for submitting your manuscript to PLOS ONE. After careful consideration, we feel that it has merit but does not fully meet PLOS ONE’s publication criteria as it currently stands. Therefore, we invite you to submit a revised version of the manuscript that addresses the points raised during the review process.

**ACADEMIC EDITOR: **

Thank you for addressing almost all the issues raised in the earlier submission. However, there are few important aspects that still need attention and they are as follows:

1. The English needs further editing. Many definite and indefinite articles are missing in the text.Avoid mixing both American and British English. Stick to one.

2. The Aim is to look at factors influencing the uptake of HPV vaccine, not HPV. Correct this.

3. Objective 2: be specific about the country (Pakistan in this case)

4. Objective 3: be specific about the community.

5. Under HPV Vaccination belief subheading, HBM should be written in full at first use.

6. Qualitative data: This data was to be collected till data saturation is achieved. However, the authors stated that only 4 FGDs will be conducted with the KIIs. Please, reconcile these to clearly show how sample size will be determined.

We look forward to receiving your revised manuscript.

Kind regards,

Folusho Mubowale Balogun

Academic Editor

PLOS ONE
---

## [Author Response · Author response to Decision Letter 1]

11 Apr 2025

Dear Editor,

We are thankful to the worthy reviewers for sparing their valuable time to review our manuscript. Please find the description and justification of point-to-point comments and line .

S.# Comment Response Reply/Changes

1 The English needs further editing. Many definite and indefinite articles are missing in the text.Avoid mixing both American and British English. Stick to one.

Document is thoroughly reviewed and edited

Throughout document

2 The Aim is to look at factors influencing the uptake of HPV vaccine, not HPV. Correct this.

Corrected Page 6

Line 186-187

3 Objective 2: be specific about the country (Pakistan in this case

Added accordingly Page 6

Line 190-191

4 Objective 3: be specific about the community Specific community setting added in objective and methodology both

Page 6 Line 192-193

Page 7 Line 225-226

5 Under HPV Vaccination belief subheading, HBM should be written in full at first use.

Added in full form Page 8

Line 271

6 Qualitative data: This data was to be collected till data saturation is achieved. However, the authors stated that only 4 FGDs will be conducted with the KIIs. Please, reconcile these to clearly show how sample size will be determined.

Sample size for qualitative data is determined by data saturation , a point where new information no longer contributes to understanding of phenomena of interest, our initial sample size are based on normal saturation points in comparable research30. Since point of saturation cannot be predicted in advance so we planed to recruit minimum of 25-30 caregivers and potential recipients , 8-10 policy makers and 25-30 healthcare providers which is in accordance with the typical points of data saturation reported in qualitative studies31. However we are willing to modify our sample size if needed, and we will continue to recruit until saturation is reached to ensure a indepth knowledge of the factors influencing HPV vaccine uptake.

Page 11

Line 370-378

---

## [Editor Report · Decision Letter 2]

16 Apr 2025

Public Health Implications of Introducing Human Papillomavirus (HPV) Vaccination in Pakistan: A protocol for mixed method study to explore community perceptions and health system preparedness

PONE-D-24-59459R2

Dear Dr. Khola Noreen,

We’re pleased to inform you that your manuscript has been judged scientifically suitable for publication and will be formally accepted for publication once it meets all outstanding technical requirements.

Kind regards,

Folusho Mubowale Balogun

Academic Editor

PLOS ONE
---

## [Editor Report · Acceptance letter]

PONE-D-24-59459R2

PLOS ONE

Dear Dr. Noreen,

I'm pleased to inform you that your manuscript has been deemed suitable for publication in PLOS ONE. Congratulations! Your manuscript is now being handed over to our production team.

Kind regards,

on behalf of

Dr. Folusho Mubowale Balogun

Academic Editor

PLOS ONE